

# Microbial communities inhabiting the surface and gleba of white (*Tuber magnatum*) and black (*Tuber macrosporum*) truffles from Russia

Ekaterina V. Malygina[1], Nadezhda A. Potapova[1], Natalia A. Imidoeva[1], Tatiana N. Vavilina[1], Alexander Yu Belyshenko[1], Maria M. Morgunova[1], Maria E. Dmitrieva[1], Victoria N. Shelkovnikova[1], Anfisa A. Vlasova[1], Olga E. Lipatova[1], Vladimir M. Zhilenkov[1], Anna A. Batalova[1], Elina E. Stoyanova[2] and Denis V. Axenov-Gribanov[1]

[1] Laboratory of Experimental Neurophysiology, Department of Research and Development, Biological Faculty, Irkutsk State University, Irkutsk, Russia
[2] East-Siberian State University, Krasnoyarsk, Russia

Corresponding authors
Ekaterina V. Malygina,
cat.malygina@gmail.com
Denis V. Axenov-Gribanov,
denis.axengri@gmail.com

## ABSTRACT

The complex symbiotic relationships between truffles and their microbiota, coupled with their obligate mycorrhizal lifestyle, present significant challenges for obtaining axenic mycelium and achieving controlled cultivation. This study aimed to characterize the microbial communities within the surface and gleba of truffle ascomata using 16S and 18S rRNA gene sequencing and identify the taxonomic composition and ecological roles of these microbiota. Specimens of *Tuber magnatum* (white truffle) and *Tuber macrosporum* (smooth black truffle) were collected, with *T. magnatum* representing the first documented discovery of this species in Russia. Metabarcoding profiling identified both species-specific and shared microbial taxa, with the yeast-like fungus *Geotrichum* spp. emerging as a core symbiont in both truffle species. Its consistent detection in surface and gleba tissues suggests a critical role in mycorrhizal establishment and spore dispersal, potentially mediated by sulfur volatiles that attract mycophagous fauna. In *T. magnatum*, the bacterial community was dominated by Proteobacteria, particularly Alphaproteobacteria and Gammaproteobacteria, with the nitrogen-fixing genus *Bradyrhizobium* being especially abundant. The truffle microbiota predominantly comprised soil-derived microorganisms (*e.g.*, nitrogen-fixing *Rhizobiaceae* spp., phenol-degrading *Mycoplana* spp.) and plant-associated symbionts (*e.g.*, ectomycorrhizal *Sebacina* spp.), implicating these communities in nutrient cycling, xenobiotic degradation, and host plant interactions. By elucidating the taxonomic and functional profiles of truffle-associated microbiota, this study provides foundational insights into their ecological contributions. Chemical differences align with tissue-specific microbial communities, suggesting microenvironmental specialization in bioactive compound synthesis. These findings advance efforts to replicate critical symbiotic interactions *in vitro*, a prerequisite for developing sustainable cultivation protocols for *T. magnatum* and *T. macrosporum* under controlled conditions.

## INTRODUCTION

Truffles are hypogeous fungi whose growth is contingent upon specific climatic and edaphic conditions (*Robin, Goutal-Pousse & Le Tacon, 2016*). These fungi predominantly thrive in temperate climates with distinct seasons, which provide the necessary temperature and moisture variations for their life cycle (*Hall et al., 2017*). Their distribution is intrinsically linked to the range of their obligate host plants (*Gryndler et al., 2017*; *Zambonelli, Iotti & Hall, 2015*). The cultivation of truffle fungi, particularly *Tuber melanosporum* Vittad. and *Tuber aestivum* Vittad., has a long-established history in Southern and Central Europe. Recent advancements in the understanding of truffle biology have led to improved control and standardization of cultivation techniques, enabling the successful cultivation of several truffle species outside their native ranges. This includes the gastronomically prized *Tuber magnatum* Pico.

Despite its high market value and successful cultivation in regions such as Africa, *T. melanosporum* remains the most widely cultivated truffle species globally. Its cultivation has expanded significantly beyond Europe, with successful introductions in Australia, Canada, Chile, China, New Zealand, South Africa, and the United States (*Yan et al., 2017*; *Lemmond et al., 2023*). In Russia, truffles are predominantly found in deciduous and mixed forests, with notable concentrations in Krasnodar, Transcaucasia, and the Tula and Oryol regions. Additional occurrences have been documented in forests near Moscow, St. Petersburg, Smolensk, and Belgorod, among other areas (*Malygina et al., 2024a*; *Vishnevsky, 2018*).

Truffles are highly valued not only for their unique flavor but also for their significant nutritional properties. The composition of their bioactive compounds varies depending on species and geographical origin. However, carbohydrates and proteins constitute the primary components, with truffles also being rich in essential minerals, dietary fiber, amino acids, fatty acids, and healthy fats (*Panche, Diwan & Chandra, 2016*).

Truffles are characterized by a high concentration of unsaturated fatty acids, particularly oleic and linoleic acids, which collectively account for over 60% of their total fatty acid profile (*Splivallo et al., 2019*; *Ori, 2019*; *Morgunova et al., 2023*). Linoleic acid, an essential fatty acid, serves as a precursor to 1-octen-3-ol, a key aromatic compound that contributes significantly to the distinctive aroma of truffles. Oleic acid, another prominent fatty acid in truffles, is associated with well-documented health benefits, including cholesterol reduction, cardiovascular protection, and potential anti-tumor activity (*Splivallo et al., 2011*). In addition to their fatty acid content, truffle fungi are known to synthesize a diverse array of bioactive compounds, such as ascorbic acid, ergosterol, phenols, flavonoids, terpenoids, and phytosterols. Among these, flavonoids are particularly noteworthy due to their antioxidant, anti-inflammatory, antimutagenic, and antitumor properties. *Kaya & Akcura (2014)* mentioned properties and biosynthetic capability rarely observed in other edible mushrooms. Analyses of three Chinese truffle species have further revealed a rich abundance of antioxidant compounds, including ascorbic acid, $\beta$-carotene, gallic acid, and rutin, which may provide protection against oxidative stress-related diseases (*Lee et al., 2020*). The biotechnological potential of truffles is substantial, with their associated

microbiota representing a promising source for the discovery of novel therapeutic agents (*Pavić et al., 2013*). Secondary metabolites produced by these microorganisms hold significant promise for applications in medicine and biotechnology, highlighting their potential for future research and development (*Leonardi et al., 2021*).

Truffles harbor a diverse and complex microbial community, comprising bacteria, yeasts, and filamentous fungi, which colonize them throughout their life cycle. While certain microorganisms, such as *Pseudomonas* spp., yeasts, and some mold fungi, contribute to post-harvest spoilage, others, including *Bacillus* spp. and *Listeria* spp., are recognized as potential pathogens. Microbial colonization occurs in both the surface (outer shell) and gleba (inner part) of the truffle, with significant variations in microbiome composition observed between these two regions. Although the roles of these microorganisms in the truffle life cycle, nutrient acquisition, and flavor development are widely acknowledged, their specific functions and interactions remain largely undetermined (*Yu et al., 2016*; *Wu, Meenu & Xu, 2021*). It is hypothesized that bacteria, such as *Pseudomona* s spp. and members of the Enterobacteriaceae family, play a role in truffle development and maturation (*Splivallo et al., 2015*). Furthermore, the microbiome associated with truffles is implicated in the biosynthesis of their characteristic volatile organic compounds (*Tejedor-Calvo et al., 2021*; *Vahdatzadeh & Splivallo, 2018*; *Vita et al., 2015*). This hypothesis is supported by experimental studies demonstrating that yeasts isolated from *T. melanosporum* and *T. magnatum* can independently produce volatile organic compounds when cultured on a medium supplemented with L-methionine. Further research has revealed that thiophene volatiles in *T. borchii* (white truffle) are generated through the biotransformation of non-volatile precursors by the associated bacterial community, rather than by the truffle itself (*Reyna & Garcia-Barreda, 2014*). A significant contribution of the bacterial community to aroma formation has also been proposed for other truffle species, including *T. melanosporum*, *T. magnatum*, and *T. aestivum* (*Hilszczańska et al., 2016*; *Buzzini et al., 2005*).

Thus, the aim of this study was to identify truffle ascomata and investigate the truffle associated microbial composition of their distinct parts, specifically the surface and gleba, through metabarcoding sequencing of 16S and 18S rRNA genes. These findings are expected to provide valuable insights into the mechanisms underlying mycorrhiza formation and ascoma development in truffles, which may contribute to the establishment of optimal conditions for their controlled cultivation.

## MATERIALS & METHODS

### Sampling

Two species of true truffles belonging to the genus *Tuber* were selected for this study: the smooth black truffle (*T. macrosporum*) and the white truffle (*T. magnatum*). The ascomata of both species exhibited a globular morphology. The surface of *T. magnatum* was white and rough, while that of *T. macrosporum* was dark brown with a pyramidal, verrucose surface. Large ascomata (>4 cm in diameter) were selected for analysis. The gleba of *T. magnatum* was grey, whereas that of *T. macrosporum* was black, both displaying a porous structure with characteristic white, branching veins (*Deveau et al., 2019*).

Truffles were collected in a pine forest near Moldovanovka village (Krasnodar region, GPS coordinates: 44.461813, 38.856708) in August 2023. The ascomata were located using trained truffle-hunting dogs and carefully excavated with a rake to ensure their integrity. The study utilized whole and intact truffle specimens for analysis. Within 1–2 h after collection, the truffles were cleaned and rinsed with running tap water using a toothbrush. Two distinct parts of the truffle ascomata were selected for metabarcoding profiling: the surface layer (surface) and the central part (gleba). The ascomata were surface-sterilized with 70% ethyl alcohol and flamed using a Bunsen burner. The surface was then completely removed using a sterile grater, and approximately 0.5–0.7 mL of surface powder was collected for DNA isolation and sequencing. The truffles were subsequently broken open around the perimeter, and an equivalent volume of gleba was excised using a sterile scalpel. Tris-EDTA (TE) buffer (10 mM Tris–HCl, 1 mM EDTA) was added to the resulting samples prior to total DNA extraction.

Additionally, one of the minor objectives of the current study was to visualize the distinct composition of natural compounds in different parts of the ascocarps of *T. macrosporum*. For this purpose, natural compounds were extracted from distinct and freshly dissected parts of the ascoma using a general liquid extraction method with acetonitrile and methanol, followed by high performance liquid chromatography–mass spectrometry (HPLC-MS) analysis. Due to limitations in our ability to provide a detailed analysis, we are able to provide only general chromatographic profiles. A more detailed methodology can be found in the description of the figures included in the additional materials (Figs. S4–S5) (*Pereliaeva et al., 2022*; *Morgunova et al., 2023*).

### DNA extraction

DNA was extracted from truffle ascomata using mechanical homogenization to disrupt the chitinous cell walls. Metal beads were added to the tubes containing the biomaterial suspended in TE buffer. The samples were homogenized using a BABR x1 vibrating grinder (Mycotech LLC, Irkutsk, Russia) at 3,000 rpm for one minute. This process was repeated three times, with cooling periods between each homogenization cycle (*Malygina et al., 2024a*; *Malygina et al., 2024b*).

Total DNA for metabarcoding profiling was extracted from crushed truffle ascomata using the DNeasy Plant Mini Kit (69106; QIAGEN) following the manufacturer's recommendations (*Tuovinen et al., 2019*). The prepared samples, containing isolated DNA, were sent to BioSpark LLC (Troitsk, Russia) *via* express post at a temperature of 4 °C for metabarcoding profiling.

### Amplification, sequencing, bioinformatic data processing

Metabarcoding profiling of truffle-associated microbial communities was performed through amplification of 16S rRNA gene fragments spanning the hypervariable V3–V4 regions using universal eubacterial primers. Fungal communities were analyzed by amplifying the internal transcribed spacer (ITS) regions, specifically targeting the ITS2 region and adjacent hypervariable regions of the 18S rRNA gene (Table 1). These primers are specific for the amplification of fungal DNA.

Malygina et al. (2025), *PeerJ*, DOI 10.7717/peerj.20037

**Table 1  A mixture of primers for metagenomic profiling of truffles by 16S rRNA and 18S rRNA genes used in this study.**

| Type of primer | Name | Sequence 5′–3′ | Concentration μM | Program of amplification |
|---|---|---|---|---|
| | | For 16S rRNA | | |
| F | GPro341F | CCTACGGGNBGCASCAG | 0.625 | |
| R | GPro806R | GGACTACNVGGGTWTCTAATCC | 0.625 | |
| F | NR_16_341F1 | TCGTCGGCAGCGTCAGATGTGTATAAGAGACA GTGCCTACGGGNBGCASCAG | 2.5 | |
| F | NR_16_341F2 | TCGTCGGCAGCGTCAGATGTGTATAAGAGACA GCTGCCTACGGGNBGCASCAG | 2.5 | |
| F | NR_16_341F3 | TCGTCGGCAGCGTCAGATGTGTATAAGAGACA GTCTGCCTACGGGNBGCASCAG | 2.5 | 95 °C–3 min (initial denaturation); 95 °C–30 s, 57 °C–30 s, 72 °C–30 s (35 cycles); 72 °C–5 min (final extension step). |
| R | NR_16_806R1 | GTCTCGTGGGCTCGGAGATGTGTATAAGAGACAGCC GGACTACNVGGGTWTCTAATCC | 2.5 | |
| R | NR_16_806R2 | GTCTCGTGGGCTCGGAGATGTGTATAAGAGACAGA CCGGACTACNVGGGTWTCTAATCC | 2.5 | |
| R | NR_16_806R3 | GTCTCGTGGGCTCGGAGATGTGTATAAGAGACAGAA CCGGACTACNVGGGTWTCTAATCC | 2.5 | |
| | | For 18S rRNA | | |
| F | NR_5.8SR | TCGTCGGCAGCGTCAGATGTGTATAAGAGACAGATCT CGATGAAGAACGCAGCG | 5.0 | 95 °C–3 min (initial denaturation); 95 °C–30 s, 55 °C–30 s, 72 °C–30 s (7 cycles); 72 °C–5 min (final extension step). |
| R | NR_ITS4R | GTCTCGTGGGCTCGGAGATGTGTATAAGAGACAGGCA TCCTCCGCTTATTGATATGC | 5.0 | |

Library preparation was conducted using the dual-index Nextera Index Kit (Illumina) following the manufacturer's protocol (*Hirose et al., 2020*). Following PCR amplification, amplicons were purified using AMPure XP magnetic beads (KAPA Biosystems) to remove primer dimers and non-specific products. Purification steps were performed in accordance with the manufacturer's guidelines and (*Mahajan & McLellan, 2020*).

Nucleotide sequences were subjected to shotgun metagenomic sequencing. High-throughput sequencing was performed on the Illumina MiSeq platform using paired-end sequencing with a minimum output of 10,000 paired-end reads per sample. Libraries were prepared and sequenced using the MiSeq Reagent Kit v2 Nano and MiSeq v2 Reagent Kit (500-cycle configuration) following the manufacturer's protocols (*Sato et al., 2019*).

Bioinformatic data processing began with checking the quality of the raw readings. The quality assessment was performed using the FastQC program (v. 0.11.9 *Andrews, 2010*). Adapters from the sequence were cut using the following parameters: HEADCROP:20, CROP:221, LEADING:3, TRAILING:3/SLIDING WINDOW:5:10, AVGQUAL:20, MINLEN:221 (*Chen et al., 2018*). The length of the nucleotide sequences was 221 nucleotides. The data was then analyzed using the DADA2 algorithm (v. 1.26.0) in R (version 4.2.1). The reads were filtered, paired, chimeras were removed, and then a search was performed on the Silva database (for 16S - silva_nr99_v138.1_train_set, for 18S -SILVA_132_SSURef_tax_silva version).

## Phylogenetic analysis

Phylogenetic analysis was conducted using MEGA X (*Kumar et al., 2018*). Nucleotide sequences were aligned with those displaying the highest similarity available in the NCBI database, utilizing the ClustalW algorithm with default parameters. Quality control check of the sequences was performed using UGENE (v. 39.0) (*Okonechnikov et al., 2012*). Sequences were aligned with the MUSCLE tool implemented in UGENE (v. 39.0) with default parameters. Pairwise distances were calculated using MEGA X with 100 bootstrap replications, uniform rates among sites, transitions and transversions were included, also we chose pairwise deletion for missing data (gaps).

The best model for phylogenetic analysis of Russian *T. magnatum* was K2 (Kimura2-parameter model) according to Bayesian information criterion (BIC) and was obtained by the Model Selection tool implemented in MEGA X with default parameters. Phylogenetic reconstruction was performed using the maximum likelihood method with 100 bootstrap replications in MEGA X. Visualization was performed with iTOL (*Letunic & Bork, 2021*). The best model for phylogenetic analysis of Russian *T. macrosporum* was T92+G according to BIC and was obtained by the Model Selection tool implemented in MEGA X with default parameters. Phylogenetic reconstruction was performed using the maximum likelihood method with 100 bootstrap replications in MEGA X. Visualization was performed with iTOL.

## Statistics

Metabarcoding analysis was performed on three ascomata of *T. macrosporum* and four ascomata of *T. magnatum*. The reliability of the data was assessed using the Mann–Whitney

*U* test. Statistical analysis was conducted using the software Past 4.10 (Natural History Museum, Oslo, Norway). Graphical representations were generated based on mean values ± standard deviations throughout the study.

### Construction of co-occurrence networks of microbial communities

We developed a comprehensive Python pipeline for analyzing truffle-associated microbial communities, covering all stages from data processing to network visualization. The analysis began with construction of a ≪ sample × taxon≫ abundance matrix using pandas and numpy libraries. Microbial abundance data underwent centered log-ratio transformation for normalization, followed by removal of rare taxa (prevalence <10%) (*Bars-Cortina, 2022*).

Statistical analysis performed with scipy.stats included calculation of Pearson correlations for all taxon pairs, with *p*-value estimation and subsequent false discovery rate correction to minimize type I errors. Statistically significant correlations ($|r| \geq 0.1$, $p \leq 0.05$) were used to construct interaction networks *via* networks library (*Ma et al., 2021*), where taxa were represented as nodes and significant correlations as edges (red for positive, blue for negative associations).

Network topology was characterized using key metrics: modularity (for detecting functional communities), clustering coefficient, node centrality (degree centrality), along with within-module connectivity (Zi) and among-module connectivity (Pi) indices. Visualization implemented with matplotlib included: taxonomic coloring of nodes, size scaling by abundance or degree centrality, edge thickness proportional to correlation strength, force-directed layout optimization, and detailed legend creation using matplotlib.lines.Line2D.

## RESULTS

### Identification and phylogenetic analysis

Several truffle species have been identified through molecular methods, including *T. macrosporum* and *T. magnatum*. Their genetic affiliation has been confirmed by phylogenetic analysis, which revealed well-defined clades consistent with other representatives of the same species. Notably, all three representative ascomata of *T. macrosporum* form a mixed clade on the phylogenetic tree, clustering with representatives of this species from different countries (Fig. S1). Additionally, this study reports the presence of *T. magnatum* Krasnodar region of Russia for the first time (Fig. S2), a species previously documented only in several European countries, such as Italy, France, and Croatia (*Bach et al., 2021*). The Russian *T. magnatum* specimens did not form independent clades and exhibited high genetic similarity with other representatives of *T. magnatum* (Fig. S3).

### Truffle associated communities in the surface and gleba of *T. macrosporum*

Metabarcoding profiling of the surface and gleba of the smooth black truffle *T. macrosporum* revealed a diverse eukaryotic community. Sequencing of the 18S rRNA gene identified a wide range of eukaryotic microorganisms within the ascoma. Significant differences in the

composition of eukaryotic microbial communities were observed between the surface and gleba (Fig. 1).

The eukaryotic community inhabiting *T. macrosporum* (OTU 88.1 ± 3.9%) also included members of the phyla Ascomycota (OTU 12.02 ± 4.46%) and Basidiomycota (OTU 3.78 ± 2.11%), and was represented by 21 distinct fungal genera.

The eukaryotic microbial community within the surface of T. macrosporum ascomata included the following genera: *Exophiala* sp. (OTU 9.5%), *Geotrichum* sp. (OTU 5.77%), *Sebacina* sp. (OTU 4.9%), *Aspergillus* sp. (OTU 0.75%), *Alternaria* sp. (OTU 0.34%), *Diutina* sp. (OTU 0.3%), *Xenasmatella* sp. (OTU 0.18%), *Cladosporium* sp. (OTU 0.16%), *Debaryomyces* sp. (OTU 0.15%), *Pluteus* sp. (OTU 0.11%), *Lodderomyces* sp. (OTU 0.08%), *Trichosporon* sp. (OTU 0.06%), *Hyphopichia* sp. (OTU 0.05%).

The minor eukaryotic groups identified in the *T. macrosporum* ascoma sample included the following genera: *Geotrichum* sp. (OTU 3.6 ± 3.16%), *Peniophora* sp. (OTU 0.46 ± 0.46%), *Cladosporium* sp. (OTU 0.27 ± 0.16%), *Diutina* sp. (OTU 0.26 ± 0.3%), *Yarrowia* sp. (OTU 0.23 ± 0.22%), *Candida* sp. (OTU 0.19 ± 0.02%), *Plectosphaerella* sp. (OTU 0.14 ± 0.14%), *Trametes* sp. (OTU 0.07 ± 0.07%). Notably, the genus *Geotrichum* sp. was detected in both the surface sample and the truffle gleba of *T. macrosporum*.

Analysis of bacterial communities inhabiting the surface and core of *T. macrosporum* ascomata revealed that the dominant classes were Alphaproteobacteria (OTU 31.01 ± 35.35%), Actinobacteria (OTU 8.22 ± 4.73%), Gammaproteobacteria (OTU 2 ± 24.92%), Bacilli (OTU 1.49 ± 20.21%), and Saccharimonadia (OTU 1.42 ± 1.23%). Minor classes, each constituting less than 1% of the community, included Clostridia, Cyanobacteriia, Planctomycetes, Bacteroidia, and Desulfurobacteriia.

The bacterial community within the surface of *T. macrosporum* ascomata comprised 27 families (Fig. 2). The dominant groups included the bacterial families *Lachnospiraceae* spp. (OTU 58.23%), *Streptococcaceae* spp. (OTU 49.75%), *Yersiniaceae* spp. (OTU 33.73), *Mitochondria* spp. (OTU 19.02 ± 32.78%), *Rhizobiaceae* spp. (OTU 14.38 ± 10.64%), *Geitlerinemaceae* spp. (OTU 12.99%), *Flavobacteriaceae* spp. (OTU 12.98), *Nocardioidacea* e spp. (OTU 10.11%). Minor groups, each constituting less than 10% of the community, included the following families: *Lactobacillaceae* spp. (OTU 5.63%), *Clostridiaceae* spp. (OTU 5.14%), *Listeriaceae* spp. (OTU 3.43%), *Xanthobacteraceae* spp. (OTU 3.39 ± 1.28%), *Mycobacteriaceae* spp. (OTU 3.37%), *Micrococcaceae* spp. (OTU 2.94%), *Saccharimonadaceae* spp. (OTU 2.26 ± 1.19%), *Dongiaceae* spp. (OTU 2.05%), *Pectobacteriaceae* spp. (OTU 1.48%), *Pirellulaceae* spp. (OTU 1.47%), *Paenibacillaceae* spp. (OTU 1.31%), *Xanthobacteraceae* spp. (OTU 1.28%), *Beggiatoaceae* spp. (OTU 1%), *Anaplasmataceae* spp. (OTU 0.71%), *Vagococcaceae* spp. (OTU 0.5%), *Desulfurobacteriaceae* spp. (OTU 0.1%).

Thus, the unique families of bacteria for surface were *Clostridiaceae* spp., *Dongiaceae* spp., *Flavobacteriaceae* spp., *Lachnospiraceae* spp., *Lactobacillaceae* spp., *Micrococcaceae* spp., *Mycobacteriaceae* spp., *Nocardioidaceae* spp., *Paenibacillaceae* spp., *Streptococcaceae* spp., *Vagococcaceae* spp. Evaluation of the metagenome of the truffle fungus *T. macrosporum* revealed several genera of bacteria found only in its core *Erysipelatoclostridiaceae* spp. (OTU

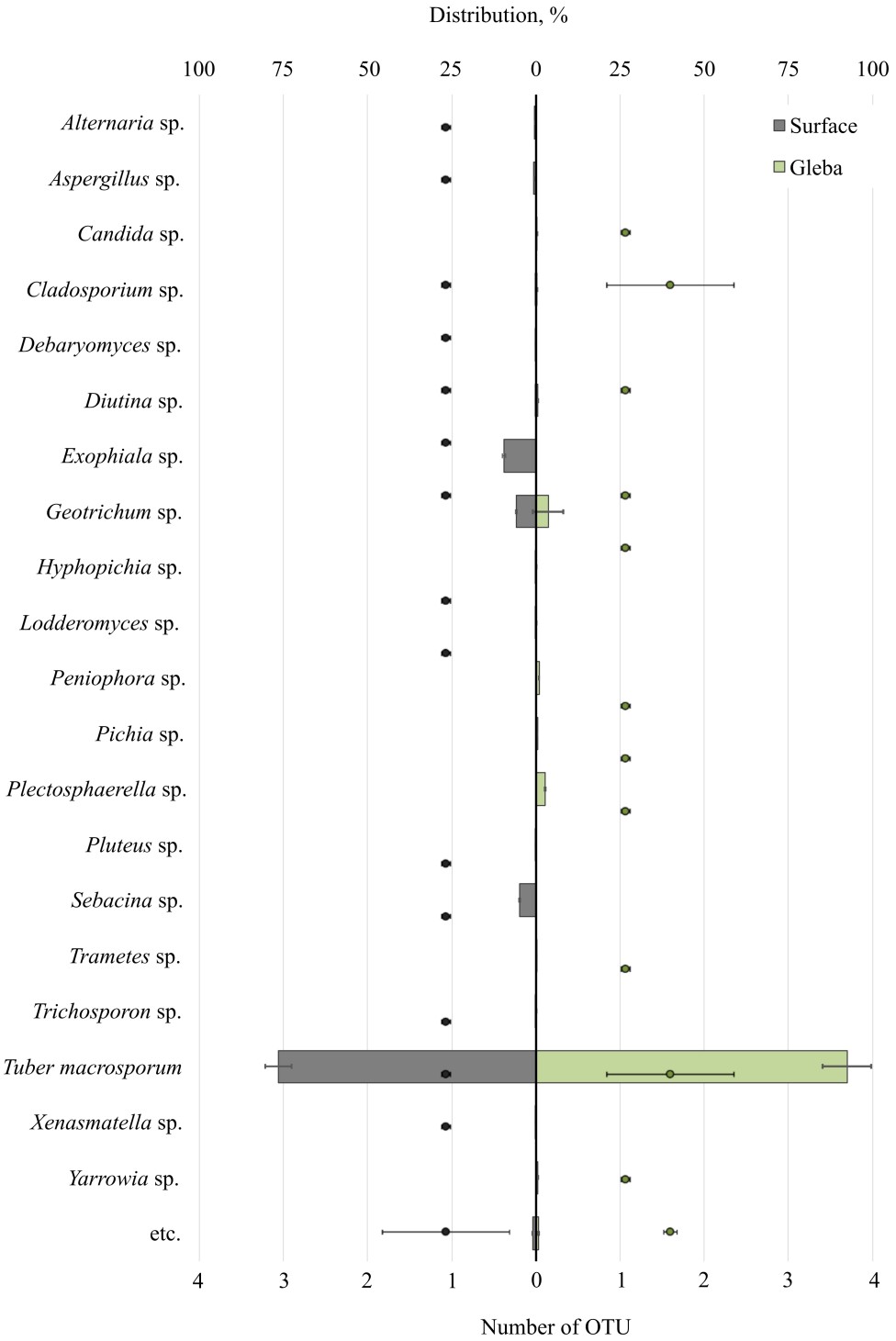

**Figure 1** **The distribution of fungal communities inhabiting the ascomata of *Tuber macrosporum* is expressed in operational taxonomic units (OTUs, %).** The histogram represents the percentage ratio (upper scale), while the dots indicate the OTU values (lower scale). Error bars (whiskers) represent confidence intervals.

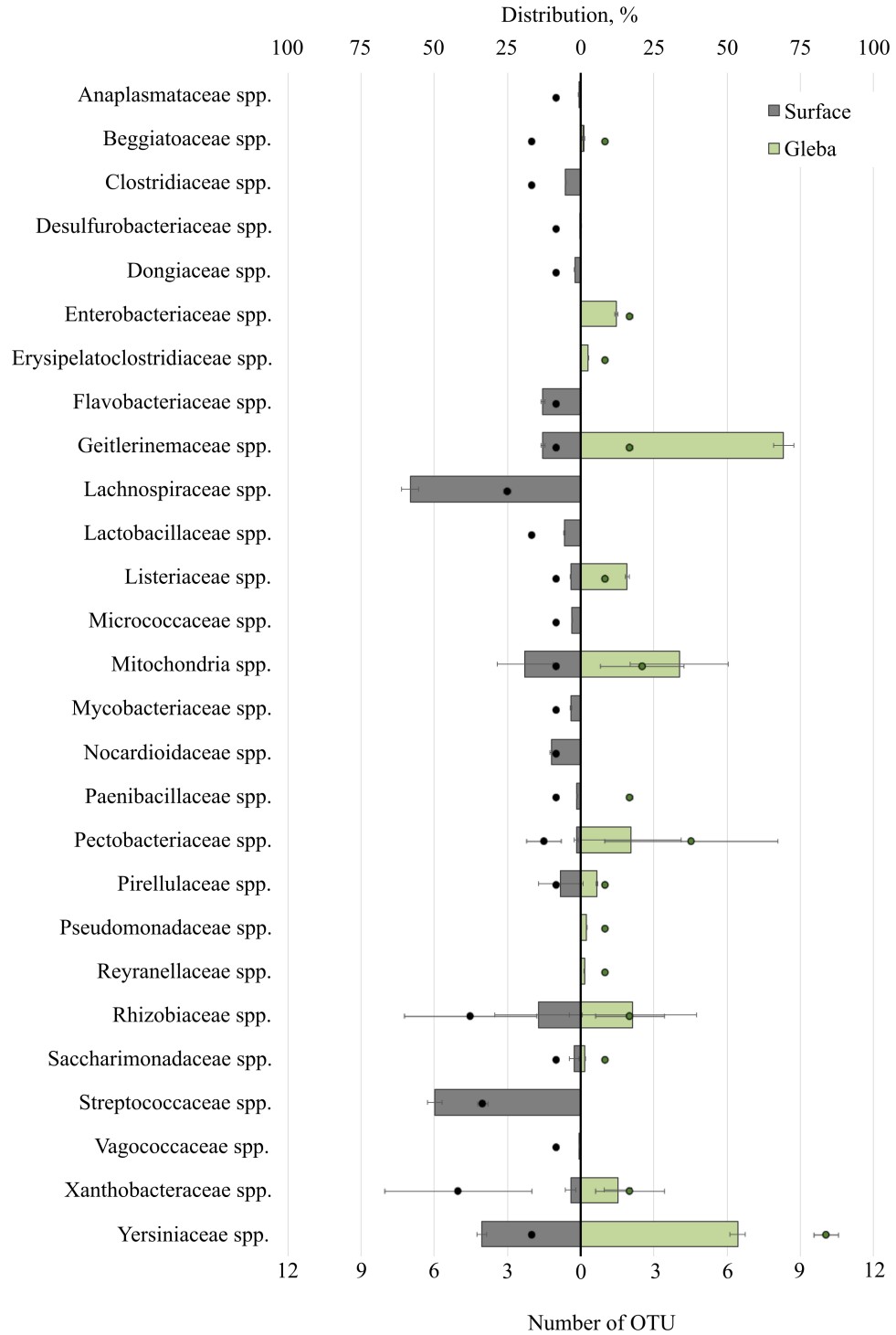

**Figure 2** **The distribution of bacterial communities inhabiting the ascomata of *Tuber macrosporum* is expressed in operational taxonomic units (OTUs, %).** The histogram represents the percentage ratio (upper scale), while the dots indicate the OTU values (lower scale). Error bars (whiskers) represent confidence intervals.

12.21%), *Pseudomonadaceae* spp. (OTU 1.87%), *Enterobacteriaceae* spp. (OTU 1.48%), *Reyranellaceae* spp. (OTU 1.24%).

## Truffle associated communities in the surface and gleba of *T. magnatum*

The community of eukaryotes inhabiting *T. magnatum* (OTU 90.71 ± 2.55%) additionally consisted of the phylum Ascomycota (OTU 8.9 ± 3.79%), Streptophyta (OTU 2.69%), Basidiomycota (OTU 0.29 ± 0.01%), Ciliophora (OTU 0.04%), and a proportion of undescribed taxa (OTU 0.08 ± 0.05%) (Fig. 3).

Twenty-five genera represented the fungal community of *T. magnatum*. The surface sample of the ascoma of *T. magnatum* contained such genera as *Geotrichum* sp. (OTU 6.84 ± 0.67%), *Diutina* sp. (OTU 0.94 ± 0.23%), *Capronia* sp. (OTU 0.74%), *Exophiala* sp. (OTU 0.61%), *Pichia* sp. (OTU 0.17%), *Rhodotorula* sp. (OTU 0.16%), *Cystolepiota* sp. (OTU 0.13%), *Geastrum* sp. (OTU 0.11%), *Phlebiopsis* sp. (OTU 0.15%), *Cladophialophora* sp. (OTU 0.1%), *Tetracladium* sp. (OTU 0.1%), *Serendipitacea* e sp. (OTU 0.09%), *Yarrowia* sp. (OTU 0.09%), *Ascobolus* sp. (OTU 0.08%), *Dactylospora* sp. (OTU 0.06%), *Alternaria* sp. (OTU 0.05%), *Hypholoma* sp. (OTU 0.04%), *Wallemia* sp. (OTU 0.03%), *Entoloma* sp. (OTU 0.01%), *Cortinarius* sp. (OTU 0.01%).

Fungi of the following genera were found in the *T. magnatum* gleba's sample: *Geotrichum* sp. (OTU 5.44 ±3.25%), *Triticum* sp. (OTU 2.69%), *Diutina* sp. (OTU 0.93 ± 0.91%), *Pichia* sp. (OTU 0.13%), *Cladosporium* sp. (OTU 0.08%), *Pelagostrobilidium* sp. (OTU 0.04%).

The *T. magnatum* truffle prokaryote community was represented by 11 families of bacteria (Fig. 4). The surface of the ascoma of *T. magnatum* included such families as *Mitochondria* spp. (OTU 55.39 ± 0.01%), *Enterobacteriaceae* spp. (OTU 15.37 ± 0.76%), *Rhizobiaceae* spp. (OTU 15.17 ± 2.07%), *Pectobacteriaceae* spp. (OTU 11.92%), *Nostocaceae* spp. (OTU 5.5 ± 0.1%), *Halobacteriaceae* spp. (OTU 4.77 ± 0.23%), *Beijerinckiaceae* sp. (OTU 3.4 ± 1.75%), *Listeriaceae* spp. (OTU 3.03%), *Clostridiaceae* spp. (OTU 0.83%), *Saccharimonadaceae* spp. (OTU 3.89 ± 2.15%).

Analysis of the surface of the ascoma of *T. magnatum* has identified unique families of *Clostridiaceae* spp., *Enterobacteriaceae* spp., and assessment of the metagenome of the truffle fungus *T. magnatum* has revealed several genera of bacteria found only in its core *Yersiniaceae* spp. and *Beijerinckiaceae* spp.

## Network analysis of truffle-associated microbial communities

This study presents a comparative network analysis of bacterial communities associated with smooth black (*T. macrosporum*) and white (*T. magnatum*) truffles, revealing distinct structural and functional patterns in their microbial and fungal consortia. The investigation employed correlation network construction and topological analysis using Zi-Pi metrics to characterize these complex microbial systems.

The fungal correlation networks of black and white truffles exhibit distinct structural patterns despite their similar composition (Fig. 5). The black truffle network consists of 23 nodes, each representing different fungal taxa, with node shapes and colors

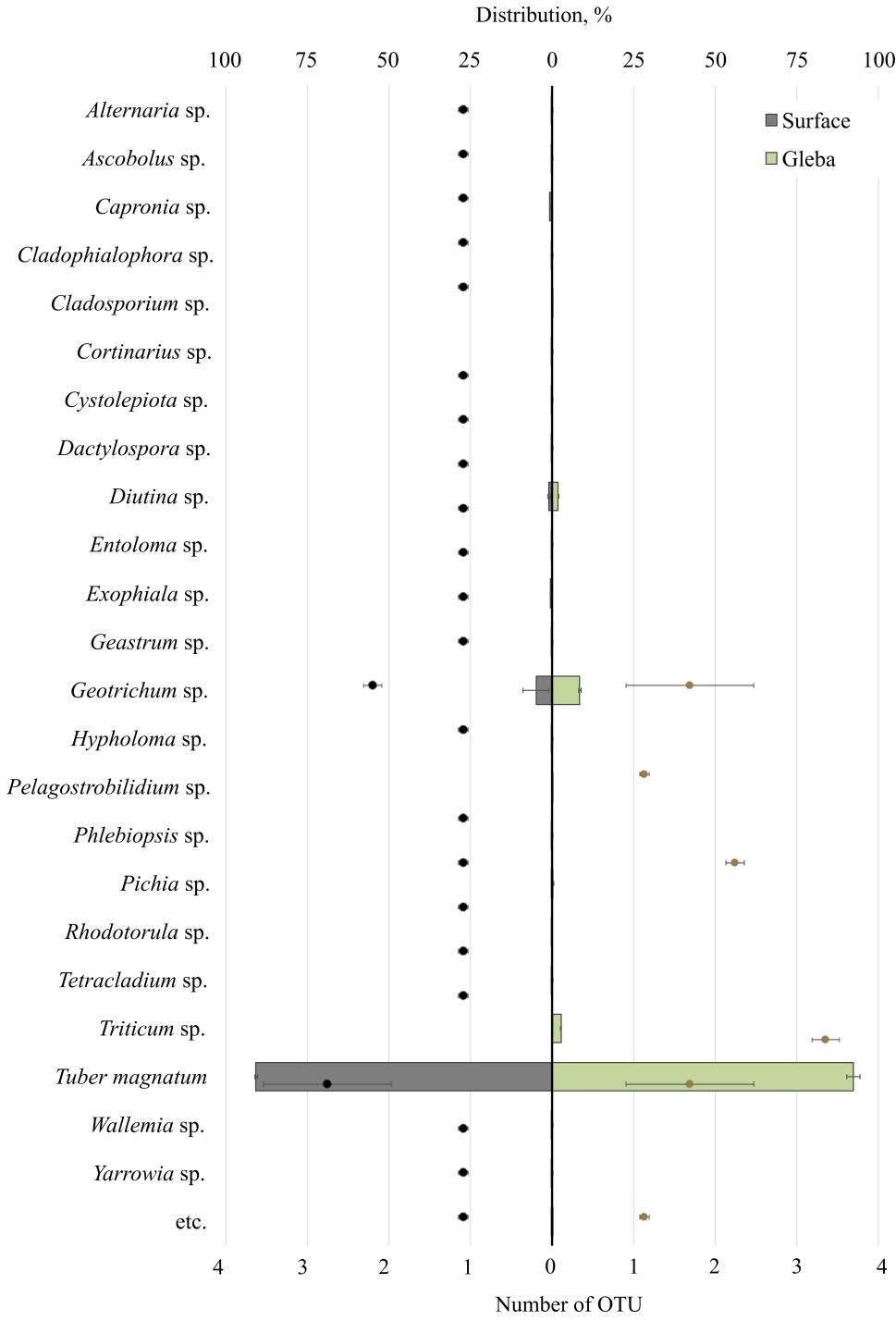

**Figure 3** **The distribution of eukaryotic microorganisms inhabiting the ascomata of *Tuber magnatum* is expressed in operational taxonomic units (OTUs, %).** The histogram represents the percentage ratio (upper scale), while the dots indicate the OTU values (lower scale). Error bars (whiskers) represent confidence intervals.

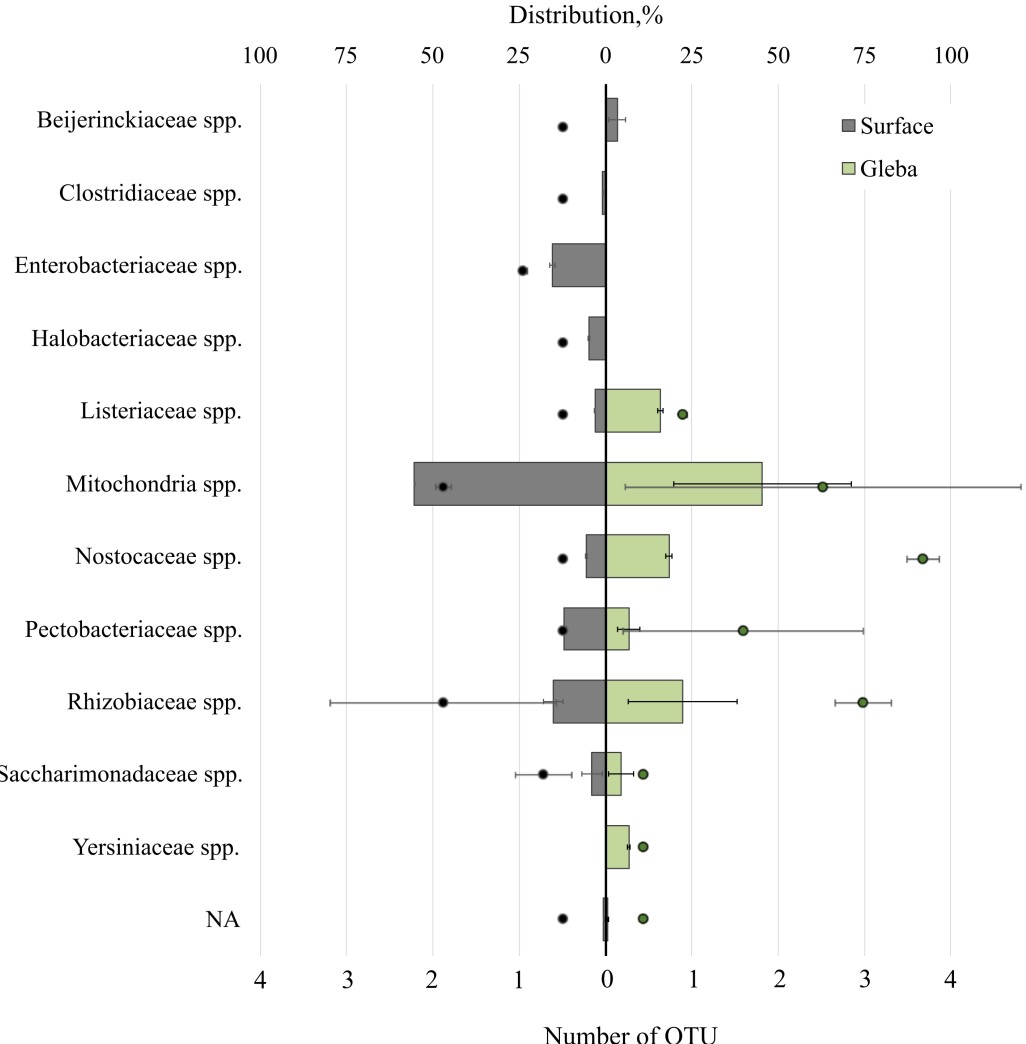

**Figure 4** **The distribution of of bacterial communities inhabiting the ascomata of *Tuber magnatum* is expressed in operational taxonomic units (OTUs, %).** The histogram represents the percentage ratio (upper scale), while the dots indicate the OTU values (lower scale). Error bars (whiskers) represent confidence intervals.

indicating phylogenetic affiliation: blue circles correspond to Ascomycota, green squares to Basidiomycota, and yellow circles to unknown taxa. Connections between nodes reflect statistically significant correlations, where red edges denote positive interactions and blue edges negative ones.

In contrast, the white truffle network displays a comparable structure with 24 nodes but differs in connection patterns and taxonomic distribution. A key distinction lies in their topological properties: while the black truffle network shows 98.8% density (indicating extremely high connectivity) and a clustering coefficient of 98.9% (suggesting strong modularity), the white truffle network reaches full connectivity (100% density) with maximum clustering (100%), forming a uniformly interconnected structure.

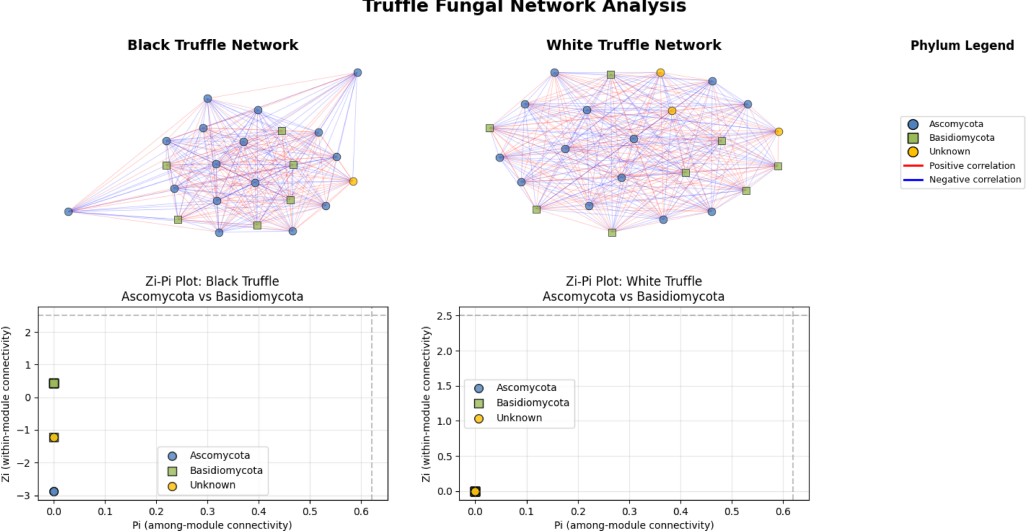

**Figure 5** Comparative analysis of fungal topology in ascomata of smooth black (*Tuber macrosporum*) and white (*Tuber magnatum*) truffles.

Additionally, the black truffle network exhibits greater structural heterogeneity, as evidenced by the Zi (within-module connectivity) values ranging from −2.87 to 0.43, pointing to differentiated roles among taxa. In contrast, the white truffle network demonstrates complete structural homogeneity, with Zi = 0 for all nodes, implying a lack of specialized hub taxa. These differences suggest that black truffles harbor a more complex and modular fungal community, whereas white truffles maintain a uniformly interconnected microbiome with fewer structural subdivisions.

Network analysis reveals divergent structural architectures between black and white truffle-associated bacterial communities, notwithstanding their taxonomic similarities (Fig. 6). The black truffle dataset comprised samples containing seven bacterial phyla, forming a network with seven significant correlations. This network exhibited high modularity (0.4552), indicating clear separation into two distinct functional modules. Notably, the majority of inter-phylum connections were positive (red edges), suggesting predominantly cooperative relationships among bacterial groups. In contrast, the white truffle network, derived from samples of similar phylum diversity (seven phyla), showed greater complexity with 8 significant correlations and lower modularity (0.1710), organized into three communities with mixed interaction types—both positive (red) and negative (blue) correlations.

Topological analysis through Zi-Pi plots revealed fundamental differences in network architecture. The black truffle network demonstrated a Zi range from −1.414 to 1.414, while all nodes showed Pi = 0, indicating complete absence of inter-module connectivity. This suggests a strictly compartmentalized structure where bacterial phyla operate within isolated functional units. Conversely, the white truffle network displayed greater topological

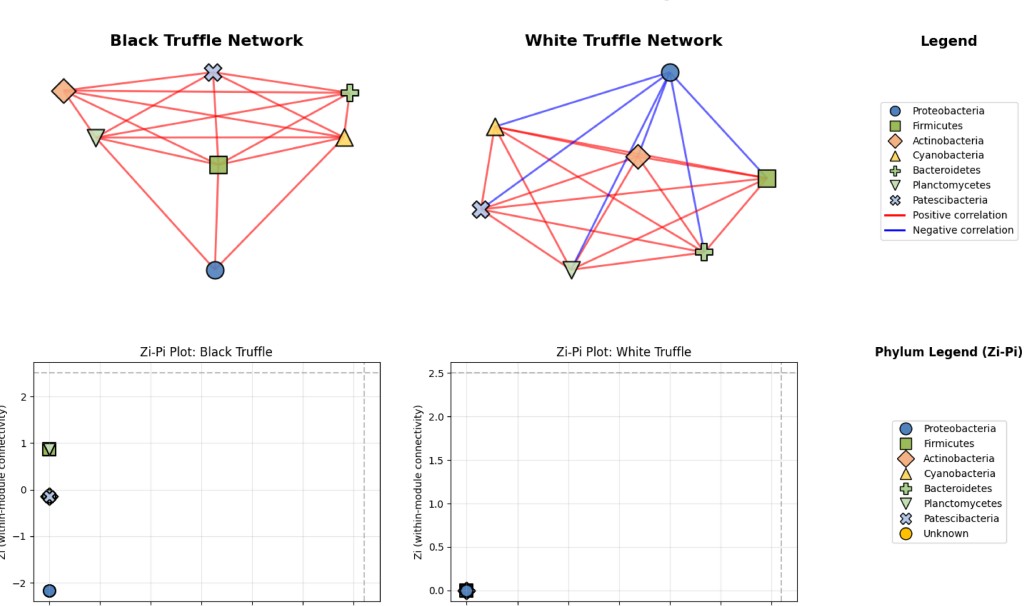

**Figure 6** Comparative analysis of bacterial community topology in ascomata of smooth black (*Tuber macrosporum*) and white (*Tuber magnatum*) truffles.

diversity with Zi values ranging from −0.707 to 1.414 and Pi values up to 0.667, confirming the presence of connector nodes that bridge different modules.

## DISCUSSION

This study investigates the microbial community composition associated with white (*T. magnatum*) and smooth black (*T. macrosporum*) truffles. Using metabarcoding profiling of the surface and gleba tissues of ascomata, we aim to identify key microorganisms associated with their development and mycorrhizal formation. Despite their culinary and economic importance, controlled cultivation of truffles remains challenging due to their obligate symbiotic relationships with specific tree hosts. Characterizing the taxonomic and functional profiles of these microbial symbionts—as well as their ecological roles—is critical for advancing cultivation strategies for these high-value fungi.

This study builds upon prior investigations of truffle-associated microbial communities in Russia. We previously characterized the microbiota of *T. aestivum* (*Malygina et al., 2024a*) and identified biochemical and chemical peculiarities (*Morgunova et al., 2023*; *Morgunova et al., 2024*) as well as bioactive properties (*Shelkovnikova et al., 2024*; *Vlasova et al., 2024*). Here we report the first confirmed discovery of the prized white truffle *T. magnatum* in Russian territory (Fig. S2), representing a significant biogeographic expansion of this species' known range. This finding is particularly noteworthy given *T. magnatum*'s traditionally restricted distribution across Southern and Central Europe, including its classic habitats in Italy, Croatia, Slovenia, and Hungary (*Belfiori et al., 2020*).

While recent studies have documented its presence in more eastern locations such as Turkey (*Doğan, Şen & Allı, 2024*) and remarkably in tropical Thailand (*Suwannarach et al., 2017*), the occurrence of this ecologically specialized truffle in Russian ecosystems presents an unexpected extension of its biogeographic boundaries. Phylogenetic analysis revealed no genetic differences between Russian specimens and conspecific populations from other regions.

Truffles (genus *Tuber*) host diverse assemblages of truffle-associated microorganisms that fulfill critical ecological roles, including facilitating mycorrhizal symbiosis establishment and ascoma development. Previous studies have characterized microbial symbionts within truffle ascomata, host plant roots, and the mycorrhizosphere, with a focus on bacterial communities associated with *Tuber aestivum*, *T. borchii*, *T. melanosporum*, *T. indicum*, and *T. magnatum* (*Antony-Babu et al., 2014*; *Barbieri et al., 2016*; *Perlińska-Lenart et al., 2020*; *Monaco et al., 2021*; *Sillo et al., 2022*; *Siebyła, Szyp-Borowska & Młodzińska, 2024*). Fungal communities associated within truffles have also been reported, though to a lesser extent (*Pacioni & Leonardi, 2016*; *Liu et al., 2020*; *Marozzi et al., 2023*). However, systematic layer-by-layer microbial profiling of *T. magnatum* and *T. macrosporum*—spanning both surface and gleba tissues—has not yet been conducted, leaving a critical gap in understanding the spatial and functional organization of their microbiota.

Performed analysis revealed that truffle-associated microbial communities are mostly presented by aerobic microorganisms. Anaerobic taxa were predominantly detected in the surface of the *T. macrosporum*. Nitrogen-fixing bacteria and producers of organic sulfur compounds were ubiquitously distributed across all ascoma tissues. In contrast, known antibiotic-producing microorganisms and organic compound synthesizers were primarily localized to the surface. Metabarcoding profiling further indicated that the microbial consortia of these truffles comprise soil-derived taxa alongside phytopathogens. Notably, animal and human-associated pathogens were also identified, with the highest relative abundance observed in surface samples (Table 2).

Metabarcoding analysis of eukaryotic communities in smooth black (*T. macrosporu*m) and white truffles (*T. magnatum*) revealed the consistent presence of *Geotrichum* spp. as a dominant fungal genus in both the surface and gleba of ascomata. Members of *Geotrichum* are implicated in the production of volatile sulfur compounds, which contribute to the characteristic aroma of truffles (*Caboni et al., 2020*). These compounds, such as dimethyl trisulfide, are derived from the enzymatic catabolism of L-methionine (*Splivallo, 2008*). For instance, the soil fungus *Geotrichum candidum* Link, 1,809 metabolizes methionine into volatile derivatives through enzymatic activity (*Bonnarme et al., 2001*; *Vahdatzadeh & Splivallo, 2018*). Such compounds act as olfactory attractants for mycophagous animals (*e.g.*, wild boars), which consume truffles and disperse their spores *via* fecal deposition. This symbiotic interaction facilitates spore dissemination and subsequent mycorrhizal colonization of host plant roots. The surface-specific enrichment of *Geotrichum* spp. suggests their potential role in mediating interactions between truffles and soil fauna. Critically, the absence of *Geotrichum* spp. in the soil microbiome may impair ascoma

Malygina et al. (2025), *PeerJ*, DOI 10.7717/peerj.20037

**Table 2** **The comprehensive distribution and functional roles of some dominant truffle associated microorganisms.**

| Kingdom | Taxon | Type of microorganism | Tuber magnatum | | Tuber macrosporum | |
|---|---|---|---|---|---|---|
| | | | Surface | Gleba | Surface | Gleba |
| Bacteria | *Beijerinckiaceae* spp. | Gram-negative, aerobes, nitrogen-fixing, free-living, methanotrophs | + | + | | |
| Bacteria | *Clostridiaceae* spp. | Gram-positive, obligate anaerobes, are part of the normoflora of the GI tract, some human and animal pathogens | + | | + | |
| Bacteria | *Enterobacteriaceae* spp. | Gram-negative, aerobes. Pathogens and producers of extended-spectrum $\beta$-lactamases, carbapenemases and L-histidine | + | | | + |
| Bacteria | *Erysipelatoclostridiaceae* spp. | Gram-positive, aerobes or facultative anaerobes, animal and human pathogens, some representatives inhabit the human gut microflora | | | | + |
| Bacteria | *Flavobacteriaceae* spp. | Gram-negative, aerobes, some representatives of facultative anaerobes, fish pathogens | | | + | |
| Bacteria | *Geitlerinemaceae* spp. | Cyanobacteria, photosynthesising. | | | + | + |
| Bacteria | *Halobacteriaceae* spp. | Archaea, most Gram-positive, mostly aerobes, extremal halophiles, free-living saprophytes | + | | | |
| Bacteria | *Listeriaceae* spp. | Gram-positive, aerobes, microaerophiles. Human and animal pathogens | + | + | + | + |
| Bacteria | *Lachnospiraceae* spp. | Gram-positive, obligate-anaerobic, inhabit the intestinal microflora of humans and animals, saprophytes (process lignocellulose and carbon dioxide), produce butyric acid | | | + | |
| Bacteria | *Lactobacillaceae* spp. | Gram-positive, facultatively anaerobic or microaerophilic, probiotics in human and animal gut microflora, produce lactic acid, participate in food fermentation | + | | | |
| Bacteria | *Micrococcaceae* spp. | Gram-positive, aerobes or facultative anaerobes, there are a small number of species classified as obligate anaerobes, saprophytes, pathogens | | | + | |
| Bacteria | *Mycobacteriaceae* spp. | Gram-positive, aerobes, acid and alcohol tolerance, saprophytes, human and animal pathogens | | | + | |
| Bacteria | *Nocardioidaceae* spp. | Gram-positive, aerobes, saprotrophs, bioindicators of gas hydrate deposits | | | + | |
| Bacteria | *Paenibacillaceae* spp. | Gram-positive, aerobes or facultative anaerobes, plant symbionts, nitrogen fixation, antibiotic producers, used as pesticides | | | + | |
| Bacteria | *Pectobacteriaceae* spp. | Gram-negative, facultative anaerobes, pectolytic. Plant pathogens | + | + | + | + |
| Bacteria | *Pirellulaceae* spp. | Gram-negative, aerobes, microaerophiles or anaerobes | | | + | + |

Malygina et al. (2025), *PeerJ*, DOI 10.7717/peerj.20037

**Table 2** (*continued*)

| Kingdom | Taxon | Type of microorganism | *Tuber magnatum* | | *Tuber macrosporum* | |
|---------|-------|----------------------|------------------|------|--------------------|------|
| | | | Surface | Gleba | Surface | Gleba |
| Bacteria | *Pseudomonadaceae* spp. | Gram-negative, aerobes, human and plant pathogens, saprotrophs, some species synthesise antibiotics and biopesticides | | | | + |
| Bacteria | *Rhizobiaceae* spp. | Gram-negative, aerobic, nitrogen-fixing. Symbiotic bacteria (symbiosis with leguminous plants). Some species are plant pathogens | + | + | + | + |
| Bacteria | *Saccharimonadaceae* spp. | Obligate epibionts (symbionts of other bacteria), possible role in the gut microbiome | + | + | + | + |
| Bacteria | *Streptococcaceae* spp. | Gram-positive, facultative anaerobes, animal and human pathogens, probiotics (used in lactase deficiency), used in the dairy industry | | | + | |
| Bacteria | *Xanthobacteraceae* spp. | Gram-negative, aerobes, plant symbionts, nitrogen fixers | | | + | |
| Bacteria | *Yersiniaceae* spp. | Gram-negative, facultative anaerobes. Human and animal pathogens | | + | + | |
| Fungi | *Aspergillus sp.* | Aerobes, saprotrophs, producers of enzymes, antibiotics, production of organic acids (citric acid, gluconic acid), human and animal pathogens | | | + | |
| Fungi | *Capronia* sp. | Aerobes, micromycetes, saprotrophs, symbionts (some form associations with lichens), human and animal pathogens, black yeasts | + | | | |
| Fungi | *Diutina* sp. | Facultative anaerobes, yeasts, human pathogens | + | | | |
| Fungi | *Exophiala* sp. | Aerobes, micro-mycetes, polyextremophilic opportunistic pathogen, black yeast | + | | + | |
| Fungi | *Geotrichum* sp. | Aerobes, saprotrophs, micromycetes, found in normal human microflora, producer of volatile organic sulphur compounds | + | + | + | + |
| Fungi | *Sebacina sp.* | Aerobes, saprotrophs, symbionts (mycorrhizae) | | | + | |
| Fungi | *Plectosphaerella* sp. | Aerobes, saprotrophs, phytopathogens, micromycetes | | | | + |

**Notes.**
"+" means presence of microorganism.

formation, as these fungi likely support truffle development through both biochemical signaling and ecological facilitation of spore dispersal.

Metabarcoding analysis revealed distinct eukaryotic communities in *T. macrosporum*, with *Exophiala* spp. and *Sebacina* spp. dominating the surface, while *Plectosphaerella* spp. and *Peniophora* spp. were characteristic of the gleba. *Exophiala* spp., known endophytes associated with *Quercus ilex* L. roots colonized by *T. melanosporum* (*Herrero de Aza et al., 2022*), and *Sebacina* spp., ectomycorrhizal fungi forming tripartite symbioses with host plants and other root-associated fungi (including Pezizales truffles) suggest roles in mediating symbiotic interactions (*Murat et al., 2008*; *Leonardi et al., 2013*; *Marjanović et al., 2020*). In contrast, *Plectosphaerella* spp. and *Peniophora* spp., phytopathogens linked to root rot in crops (*Carlucci et al., 2012*; *Lambevska, Rusevska & Karadelev, 2013*), likely represent transient colonizers. Notably, *Peniophora cinerea* (Pers.), detected in *T. borchii* ascomata, was tested for mycorrhizal involvement *via* co-cultivation with *Populus alb* a seedlings and *T. borchii* mycelium. Histological analysis using deoxynucleotidyl transferase labeling revealed no apoptosis, tannin deposition, or detectable hyphal colonization in roots, indicating no functional role in mycorrhization (*Ragnelli et al., 2014*; *Pacioni & Leonardi, 2016*). Also, metabarcoding profiling identified *Alternaria* spp., *Ascobolus* spp., *Wallemia* spp., and *Yarrowia* spp. as dominant eukaryotic taxa in the surface of *T. magnatum*. Like *Exophiala* spp., *Alternaria* spp. are endophytes previously detected on *Quercus* spp. roots colonized by *T. melanosporum* (*Herrero de Aza et al., 2022*). *Ascobolus* spp., coprophilous fungi dependent on herbivore-mediated spore dispersal, derive nutrients from undigested plant matter in ruminant manure (*Miyunga, 2015*). *Wallemia* spp., first identified in soils surrounding the Chinese truffle *Tuber indicum* (*Li et al., 2018*), and *Yarrowi* a spp., soil-dwelling yeasts known to synthesize volatile organic sulfur compounds, are linked to the distinct aroma profile of *T. magnatum* (*Splivallo, 2008*).

Despite the use of primers specific for fungal DNA in this study, we were able to obtain sequences from representatives of the *Triticum* genus (family Poaceae) from the gleba of *T. magnatum*. The genus *Triticum* spp. combines species related to cereal plants. This observation likely reflects extracellular DNA incorporation during truffle development rather than a true symbiotic association. As discomycetes, truffles develop ascomata entirely underground, where they interact with and encapsulate organic and inorganic soil components. During maturation, soil-derived genetic material—including plant DNA from nearby flora—may passively integrate into the gleba matrix. In this case, *Triticum* spp. DNA likely originated from cereal plants growing in the forest ecosystem adjacent to the truffle's habitat. While such incidental DNA uptake is a documented artifact in soil metagenomes, it underscores the importance of rigorous contamination controls when interpreting environmental sequencing data.

Metabarcoding profiling revealed that *T. magnatum* (white truffle) and *T. macrosporum* (smooth black truffle) share dominant prokaryotic taxa, including *Rhizobiaceae* (Alphaproteobacteria; *Phyllobacterium*, *Rhizobium*, *Mycoplana*), *Yersiniaceae* (Gammaproteobacteria), and *Rickettsiales* (Alphaproteobacteria), ubiquitously distributed across surface and gleba tissues. While *Rickettsiales* sequences were detected, their unresolved genus-level taxonomy complicates ecological interpretation. *Rhizobium cf. leguminosarum*

uniquely induced hyphal vacuolization in *T. aestivu* m co-cultures, whereas *Phyllobacterium* showed no interaction, and *Mycoplana*—known for phenol biodegradation—may assist in degrading soil phenanthrenes to support ascoma development (*Gryndler & Hrselova, 2012*; *Barbieri et al., 2016*). The nitrogen-fixing capacity of Rhizobiaceae suggests a role in host plant symbiosis, while Geitlerinemataceae (cyanobacteria), dominant in *T. macrosporum*, likely enhance soil nitrogen cycling as biofertilizers (*Rubin-Blum et al., 2024*). Furthermore, metabarcoding of the *T. macrosporum* gleba revealed the presence of the family Pseudomonadaceae (Gammaproteobacteria), specifically bacteria of the genus *Pseudomonas* spp. Members of the *Pseudomonas* genus are frequently detected within truffle ascomata and play significant ecological roles. Microbes associated with the truffle ascoma are involved in its developmental processes (*Chen et al., 2019*). Research by *Ballestra et al. (2010)* has also implicated this bacterial genus in the spoilage of black truffles, even during extended low-temperature storage.

In this study, we also present some initial data, visualizing the mass spectrometric profiles characterizing the natural product content of the surface and gleba of the truffle *T. macrosporum* (Fig. S4–S5). In addition to the distinct microbial composition of different parts of the ascocarps, we detected differences in the natural product composition in distant parts of the ascomata. These differences were observed under varying extraction protocols and solvents.

Thus, metabarcoding profiling of the Russian truffles *T. macrosporum* and *T. magnatum* identified dominant bacterial and fungal taxa, including soil-derived microorganisms, plant symbionts, and phytopathogens. These microbial communities appear to facilitate key stages of truffle development, such as nutrient acquisition and mycorrhizal colonization. For instance, nitrogen-fixing bacteria (*Rhizobiaceae* spp.), phenol-degrading taxa (*Mycoplana* spp.), and cyanobacterial biofertilizers (*Geitlerinemataceae* spp.) likely enhance nutrient cycling in the rhizosphere, while fungal symbionts (*Sebacina* spp., *Exophiala* spp.) mediate interactions with host plant roots. Notably, the majority of identified microorganisms may contribute to enzymatic degradation of root cell wall components, enabling truffles to establish symbiotic interfaces for nutrient exchange. This functional synergy underscores the ecological interdependence between truffles and their microbiota, which collectively support fungal proliferation in soil ecosystems.

Our study reveals striking differences in microbial network architectures between white (*T. magnatum*) and smooth black truffles (*T. macrosporum*), providing new insights into their distinct ecological strategies. The *T. magnatum* microbiome forms a completely connected network (100% density), indicating exceptionally stable and comprehensive fungal associations that likely contribute to its renowned ecological specificity. In contrast, *T. macrosporum* maintains a slightly less dense network (98.8% density) but with greater structural complexity, evidenced by variable within-module connectivity (Zi range: −2.87 to 0.43) that suggests more specialized microbial partnerships (*Antony-Babu et al., 2014*; *Monaco et al., 2022*).

Both species maintain single, non-modular communities typical of closely related fungal consortia in truffle ascomata, yet they achieve this through different topological configurations. While *T. magnatum* shows remarkable homogeneity in node roles

(all Zi = 0), *T. macrosporum* exhibits niche differentiation among microbial partners, potentially reflecting adaptation to more variable environmental conditions. These architectural differences likely stem from species-specific evolutionary pressures, including distinct habitat requirements, divergent host plant interactions, unique metabolic constraints, and varying ecological niches (*Splivallo et al., 2015*).

A particularly significant finding is the exclusive presence of bacterial connector taxa in *T. magnatum*, which appear to facilitate cross-species communication and metabolic integration—a feature conspicuously absents in *T. macrosporum*. This suggests *T. magnatum* has evolved greater dependence on microbial mediation for nutrient acquisition and cycling, possibly explaining its more restricted geographic distribution and habitat specificity compared to *T. macrosporum*.

The ecological and practical implications of these findings are substantial. For truffle cultivation, our results indicate that *T. magnatum* requires microbiome management strategies focused on maintaining network stability, while *T. macrosporum* may benefit from approaches that preserve its specialized microbial interactions. These findings demonstrating that phylogenetic proximity does not necessarily predict microbial community structure. The distinct network topologies we observed highlight the importance of considering species-specific microbial association patterns in truffle ecophysiology research and underscore the value of network analysis for developing targeted cultivation techniques that respect each species' unique microbial ecology.

## CONCLUSIONS

Consequently, this study reports the first documented discovery and characterization of *T. magnatum* in Russia. Metabarcoding profiling of *T. magnatum* and *T. macrosporum* revealed both species-specific and shared microbial taxa, enabling predictions about their functional and chemical roles in truffle biology. Notably, *Geotrichum* spp. emerged as a putative symbiotic partner common to both species, detected in both surface and gleba tissues. This ubiquitous distribution suggests a critical role in mycorrhizal symbiosis establishment and spore dispersal, potentially mediated by volatile sulfur compounds that attract mycophagous animals. In *T. magnatum*, the bacterial community was dominated by Proteobacteria, particularly Alphaproteobacteria and Gammaproteobacteria, with the nitrogen-fixing genus *Bradyrhizobium* being especially abundant.

The truffle-associated microbiome predominantly comprises soil-derived microorganisms and plant symbionts, including nitrogen-fixing bacteria (*Rhizobiaceae* spp.), phenol-degrading taxa (*Mycoplana* spp.), and cyanobacterial biofertilizers (*Geitlerinemataceae* spp.). These communities likely facilitate nutrient acquisition, organic compound degradation, and soil fertility enhancement—processes essential for ascoma development. By delineating the taxonomic and functional profiles of these microbiota, this work advances understanding of truffle ecology and provides actionable insights for optimizing cultivation strategies under controlled conditions.

## ACKNOWLEDGEMENTS

We thank all the researchers and students of the biological and soils faculty at ISU, the director of the botanical garden at ISU, and the administration of ISU for their support, and Innovation Centre Skolkovo for technical help and translation of the potential of innovation of this project.

### Funding

This work was financially supported by the Russian Science Foundation within the grant 22-76-10036. The funders had no role in study design, data collection and analysis, decision to publish, or preparation of the manuscript.

### Grant Disclosures

The following grant information was disclosed by the authors:
Russian Science Foundation: 22-76-10036.

### Competing Interests

The authors declare there are no competing interests.

### Author Contributions

- Ekaterina V. Malygina conceived and designed the experiments, performed the experiments, analyzed the data, prepared figures and/or tables, authored or reviewed drafts of the article, and approved the final draft.
- Nadezhda A. Potapova performed the experiments, analyzed the data, prepared figures and/or tables, authored or reviewed drafts of the article, and approved the final draft.
- Natalia A. Imidoeva performed the experiments, analyzed the data, prepared figures and/or tables, and approved the final draft.
- Tatiana N. Vavilina performed the experiments, analyzed the data, prepared figures and/or tables, and approved the final draft.
- Alexander Yu Belyshenko performed the experiments, analyzed the data, prepared figures and/or tables, and approved the final draft.
- Maria M. Morgunova performed the experiments, analyzed the data, prepared figures and/or tables, and approved the final draft.
- Maria E. Dmitrieva performed the experiments, analyzed the data, prepared figures and/or tables, and approved the final draft.
- Victoria N. Shelkovnikova performed the experiments, analyzed the data, prepared figures and/or tables, and approved the final draft.
- Anfisa A. Vlasova performed the experiments, analyzed the data, prepared figures and/or tables, and approved the final draft.
- Olga E. Lipatova performed the experiments, analyzed the data, prepared figures and/or tables, and approved the final draft.

- Vladimir M. Zhilenkov performed the experiments, analyzed the data, prepared figures and/or tables, and approved the final draft.
- Anna A. Batalova performed the experiments, analyzed the data, prepared figures and/or tables, and approved the final draft.
- Elina E. Stoyanova performed the experiments, analyzed the data, prepared figures and/or tables, and approved the final draft.
- Denis V. Axenov-Gribanov conceived and designed the experiments, performed the experiments, analyzed the data, prepared figures and/or tables, authored or reviewed drafts of the article, and approved the final draft.

### DNA Deposition

The following information was supplied regarding the deposition of DNA sequences:

The initial sequencing data is available in the Supplemental Files and at GenBank: PRJNA1234365 and PRJNA1234839.

The nucleotide sequences of the truffle samples studied are available at GenBank: PV212359–PV212362 and PV212363–PV212365.

### Data Availability

The initial data from metagenomic sequencing and mass spectrometry is available in the Supplemental Files.

### Supplemental Information

Supplemental information for this article can be found online at http://dx.doi.org/10.7717/peerj.20037#supplemental-information.

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
