# Peer review of "Microbial communities inhabiting the surface and gleba of white (Tuber magnatum) and black (Tuber macrosporum) truffles from Russia"

_PeerJ, doi:10.7717/peerj.20037_

## Round 0.1 · original submission · Major Revisions

·

Basic reporting

The manuscript presents data of considerable interest in the ecology of two commercially important truffles found in Russia; in fact, the microbiome of the two truffles is studied by mass sequencing with NGS, and a well-documented hypothesis on the role of these microorganisms in the metabolism of the fruiting bodies of the two truffle species is also proposed. In the context, the production of Tuber magnatum is also reported for the first time in Russia.

The work has significant scientific value and greatly expands knowledge about the white truffle, which is among the most expensive foods in the world, so it would merit its publication after review, however.

Details:
- Throughout the text and the bibliography, there are many species binomials and genus names that should be written in italics;

- Tuber macrosporum is known as “smooth black truffle” or “large-spored truffle”; the “black truffle” par excellence is Tuber melanosporum Vittad.

- The detected presence of Triticum in the metagenome deserves more attention, either because the primer used is not so specific for plants or because the truffle ascoma has probably incorporated a root fragment. This aspect should be clarified better, and if the presence of Triticum were eliminated, replace the term "Eukaryota" with Fungi.

- The peridium of Tuber magnatum is often at most 500 µm, and that of T. macrosporum is 300 µm with a greater with teeth, even 1 mm high, usually they are much higher, how can we say that the microbiome belongs only to the peridium? There is a peridium, but there is also a lot of gleba.

- The natural distribution of Tuber magnatum in the Northern Hemisphere is much wider, see:
Dogan 2024. DOI: 10.23902/trkjnat.1475517
Suwannarach,2017. https://doi.org/10.5248/132.635.

-LL 609-611 double quote delete one of the two.

Experimental design

Not perfectly correct

- The primer used is specific for fungi, not for eukaryotes.

- The peridium of Tuber magnatum is often at most 500 µm, and that of T. macrosporum is 300 µm with a greater with teeth, even 1 mm high, usually they are much higher. How can we say that the microbiome belongs only to the peridium? There is a peridium, but there is also a lot of gleba

Validity of the findings

-

Reviewer 2 ·

Basic reporting

-

Experimental design

-

Validity of the findings

The co-occurrence network of the microbial communities in different truffles should be added to the results part.

Additional comments

Inhabiting microbes such as Pseudomonas in Pezizales macrofungi (typically, truffles and morels) should be included.

Reviewer 3 ·

Basic reporting

The English language is generally appropriate. Nevertheless, not all the results related to the experiment presented in the Materials and Methods sections seem to be reported by the authors. Please check this in the text and see my comments below.

Experimental design

In general, the procedures were well described; nevertheless, in my opinion, some parts need to be detailed. Please check this in the text and see my comments below.

Validity of the findings

The findings are generally interesting, especially for T. macrosporum, as its microbiota has not been extensively studied until now. In contrast, the microbial community of T. magnatum has been more thoroughly investigated in the literature. Overall, the paper presents some intriguing results that are certainly worth sharing with the scientific community, such as the first report of T. magnatum in Russia. However, in my opinion, the findings related to truffle microbiota, while interesting, do not constitute a revolutionary novelty.

Additional comments

Please see my report below, organized section by section.

ABSTRACT

- Line 44: Correct “in vitro” with italic “in vitro”

INTRODUCTION

- Lines 49 – 50: Are you sure that this sentence is reported in the paper cited? Usually, temperate climates have high seasonal fluctuations in temperature and humidity, in contrast to tropical climates.
- Line 53: Check throughout the text that, the first time a species is mentioned, its full name is provided without abbreviation, along with the names of the species' authors (e.g., see Index Fungorum or Mycobank for fungi). Moreover, check that species names are always in italics. From the second mention onwards, you can use the abbreviation, so Tuber melanosporum Vittad becomes T. melanosporum.
- Lines 90 – 110: I suggest adding at least one sentence about the bacterial community composition of Tuber magnatum. Proteobacteria, particularly the classes Alphaproteobacteria and Gammaproteobacteria, and especially the genus Bradyrhizobium, are the most abundant bacteria found in T. magnatum. Additionally, consider mentioning the potential role of these bacteria in nitrogen fixation.

MATERIAL AND METHODS

- Also, in this section, check that species names are always in italics.
- Lines 139 – 140: I suggest using “natural compounds” instead of “natural products.”
- Line 132: In this case, it is more appropriate to refer to 'metabarcoding analyses' rather than 'metagenomic', since only two genomic regions were examined, rather than the entire genomes of the microorganisms found within the ascomata. Moreover, it is more scientifically appropriate to use the terms “ascoma” (singular) and “ascomata” (plural) instead of “fruiting body” and “fruiting bodies”. Please check and apply these corrections throughout the text.
- Lines 139 – 145: I do not see any related paragraph in the Results section, other than the graphs in the Supplementary Information. Are you sure this analysis is included in the paper? Do you really need to add it? In that case, I suggest including more details about the experimental procedure for this analysis.

RESULTS

- Lines 273 – 301 I can’t find the results on the microbial community in the gleba tissue of T. magnatum, where Bradyrhizobium is expected to be one of the most abundant genera. Please check this.
- Lines 303 – 304 After the first mention of a species name, it is no longer necessary to repeat the full name; the abbreviation can be used instead, for example, Tuber magnatum becomes T. magnatum (in italics).

DISCUSSION

- Line 320: Tuber should be in italics.
- Line 325: In my opinion, I suggest removing “sporadically”, the bacterial community of T. magnatum has been investigated in plenty of papers.
- Line 347: Please add the author names of the species Geotrichum candidum.
- Line 356: Please use the abbreviation T. macrosporum for Tuber macrosporum here as well.
- Line 359: Please add the author names of the species Quercus ilex.
- Check throughout the text that, the first time a species is mentioned, its full name is provided without abbreviation, along with the names of the species authors (e.g., see Index Fungorum or Mycobank for fungi). Moreover, check that species names are always in italics. From the second time, you can use an abbreviation, so Tuber melanosporum Vittad becomes T. melanosporum

FIGURES AND TABLES

Please remember that each figure and table caption is considered a separate text from the rest of the manuscript. Therefore, the first time a species is mentioned, its full name should be provided without abbreviation, along with the names of the species' authors (e.g., see Index Fungorum or MycoBank for fungi). Additionally, ensure that species names are always italicized. From the second mention onward, abbreviations can be used, for example, Tuber melanosporum Vittad becomes T. melanosporum.

- Figure captions: Why do you use the terms “prokaryotic microorganisms” and “eukaryotic microorganisms” instead of, for instance, “bacterial community” and “fungal community”? If all of them are respectively bacteria or fungi, I think it would be more precise to use that terminology.
- Table 1: In this case, it is more appropriate to refer to “metabarcoding analyses” rather than 'metagenomic', since only two genomic regions were examined, rather than the entire genomes of the microorganisms found within the ascomata.

---

## Round 0.2 · accepted · Accept

Thanks for addressing all comments!

·

Basic reporting

The manuscript has been revised to address the issues raised in the first revision, appropriately modifying the indication of the surface part of Tuber ascomas. Therefore, the manuscript may be published.

I would suggest changing the title by adding some specifics: "Microbial communities inhabiting the surface and gleba of white (Tuber magnatum) and black (T. macrosporum) truffles from Russia.

Experimental design

-

Validity of the findings

It adds new data to the microbiome of some true truffles (Tuber), which is then confirmation of the soil where the ascomas develop

Reviewer 2 ·

Basic reporting

-

Experimental design

-

Validity of the findings

-

Reviewer 3 ·

Basic reporting

The authors have adequately addressed all of my comments. I believe the manuscript is now ready for publication.

Experimental design

-

Validity of the findings

-